# New insights into the design of conjugated polymers for intramolecular singlet fission

Jiahua Hu[1], Ke Xu[1], Lei Shen[1], Qin Wu [2], Guiying He[1], Jie-Yu Wang[3], Jian Pei[3], Jianlong Xia [1] & Matthew Y. Sfeir [1,2]

Singlet fission (SF), a multiple exciton generation process that generates two triplet excitons after the absorption of one photon, can potentially enable more efficient solar cell designs by harvesting energy normally lost as heat. While low-bandgap conjugated polymers are highly promising candidates for efficient SF-based solar cells, few polymer materials capable of SF have been reported because the SF process in polymer chains is poorly understood. Using transient spectroscopy, we demonstrate a new, highly efficient (triplet yield of 160–200%) isoindigo-based donor–acceptor polymer and show that the triplet pairs are directly emissive and exhibit a time-dependent energy evolution. Importantly, aggregation in poor solvents and in films significantly lowers the singlet energy, suppressing triplet formation because the energy conservation criterion is no longer met. These results suggest a new design rule for developing intramolecular SF capable low-bandgap conjugated polymers, whereby inter-chain interactions must be carefully engineered.

[1] School of Chemistry, Chemical Engineering and Life Science, State Key Laboratory of Advanced Technology for Materials Synthesis and Processing, Wuhan University of Technology, 430070 Wuhan, China. [2] Center for Functional Nanomaterials, Brookhaven National Laboratory, Upton, NY 11973, USA. [3] Beijing National Laboratory for Molecular Sciences (BNLMS), The Key Laboratory of Bioorganic Chemistry and Molecular Engineering of Ministry of Education; Key Laboratory of Polymer Chemistry and Physics of Ministry of Education; Center of Soft Matter Science and Engineering; College of Chemistry and Molecular Engineering, Peking University, Beijing 100871, China. Correspondence and requests for materials should be addressed to J.X. (email: jlxia@whut.edu.cn) or to M.Y.S. (email: msfeir@bnl.gov)

Singlet fission (SF), which splits a photoexcited singlet exciton into two triplet excitons, has been suggested as a promising strategy to overcome the Shockley–Queisser limit on power conversion efficiency (PCE) of single junction solar cells[1,2]. While significant effort has been devoted in understanding the underlying mechanism of SF in molecular crystals, nanoparticles, and intramolecular compounds[3,4], the scope of materials capable of SF still remains rather limited[5]. The building blocks of SF are mainly restricted to several families of molecular chromophores, such as diphenylisobenzofurans[6,7], carotenoids[8,9], polyacenes, and their derivatives[10–14], with rare successes in polymeric materials[15–17]. However, low-bandgap conjugated polymers could be the most promising candidate for implementing high-efficiency SF-based solar cells, as they can be easily processed into thin films with good phase separation and exhibit efficient charge transport both along and between the polymer chains[18]. Donor–acceptor (D–A) type low-bandgap polymers have been widely used for organic solar cells, and have very recently improved the PCE of organic solar cells up to over 13%[19]. The use of SF-capable D–A polymers in solar cells could further boost the PCEs.

Recently, we demonstrated a design strategy for charge transfer-mediated intramolecular SF (iSF), which showed that the intra-chain D–A interactions can serve as a charge transfer intermediate for triplet pair generation[17]. We expect this process to be widespread as it has been shown that PTB1, one of the benchmark D–A polymers for organic solar cells, can generate triplet excitons via iSF in a yield of 12% upon photoexcitation[16]. Still, despite being the first reported iSF system with yields in the multiple exciton generation regime (>100%)[17], the discovery of new SF polymers has lagged far behind the small molecule iSF systems[13,20–25]. The reason that polymer systems have not experienced this same rapid growth is poor understanding of the iSF process in polymer chains. Early suggestions were that the iSF process was highly sensitive to the degree of charge transfer character[17], with later refinements suggesting that broken spatial symmetry is also required[26]. In polymers with too little charge transfer character, the singlet state decays without forming triplets, but too much CT character was shown to induce dark states that are parasitic to SF. These results show that iSF in polymers requires a balance of electronic factors that is difficult to achieve. In addition, SF within these few conjugated polymers has only been observed as an intramolecular process, with no efficient intermolecular analog to molecular crystals. As such, it is unknown how polymer SF evolves in high concentration solution and in films with different morphology[18].

In this paper, we describe the observation of morphology-dependent iSF in an isoindigo-based D–A polymer (IIDDT-Me[27], Fig. 1a), in which aggregation dramatically decreases SF. From these results, we propose an additional design rule for efficient SF polymers that requires careful engineering of inter-chain interactions and shows that aggregation can prevent the iSF from occurring in low-bandgap conjugated polymers. This requirement provides an alternative explanation for the slow pace of developing intramolecular SF polymers, since, due to their long conjugation, polymers are much more prone to aggregation than small molecules. The lack of SF in strongly aggregated systems can be explained by the shifting of absorption to low energy (preventing energy conservation from being satisfied) and the kinetic competition from inter-chain exciton dissociation. Furthermore, we identify another unique aspect of triplet pairs in polymers compared with small molecules. We observe that the driving force for SF is not static, but rather evolves in time, as structural

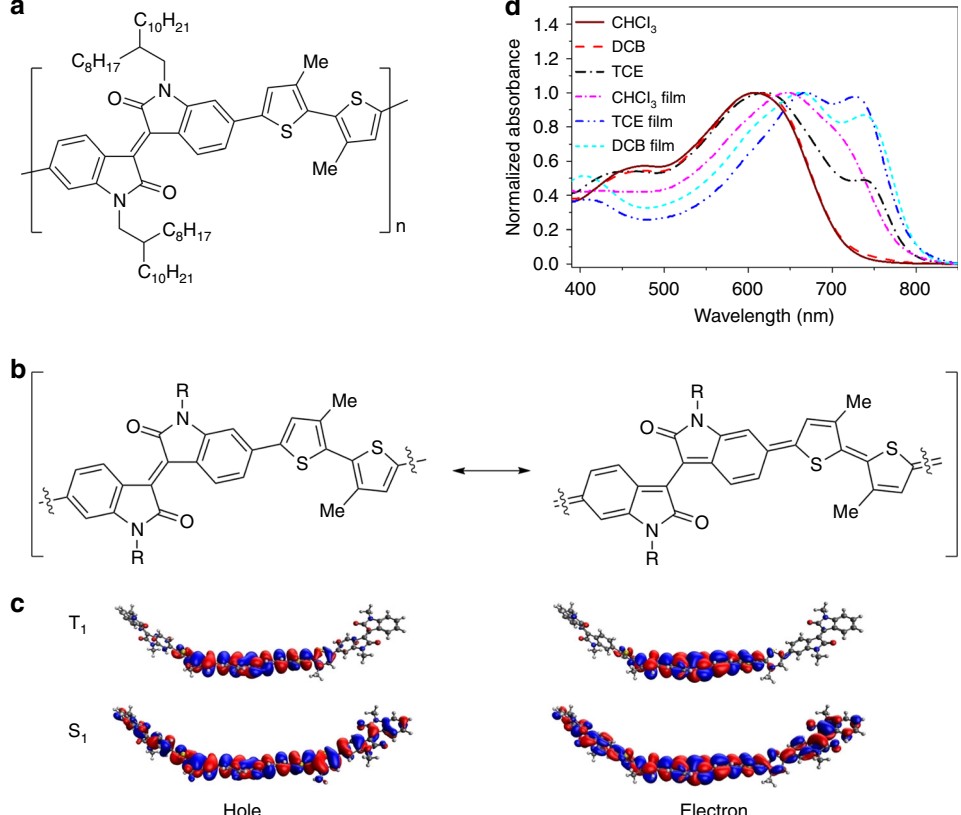

**Fig. 1** Structure and energetics analysis. **a** Structure of isoindigo-based low-bandgap polymer IIDDT-Me. **b** Aromatic and quinoidal resonance structures of IIDDT-Me. **c** Natural transition orbital analysis of the electron and hole states for the singlet and triplet. **d** Normalized UV−Vis absorption spectra of IIDDT-Me in the TCE, DCB, CHCl₃ solution, and films

relaxation changes the energy of the triplet pair relative to the singlet.

## Results

**Structure and energetics analysis.** The isoindigo-based polymer, IIDDT-Me ($M_n = 20.1$ kDa, $Đ = 3.9$), was synthesized by copolymerization of electron-deficient isoindigo chromophore and electron-rich bithiophene unit[27], and has the chemical structure depicted in Fig. 1a. The polymer chains are well beyond the oligomer regime as evidenced by high molecular weight and previous study has shown that the polymer chains exhibit a relatively narrow molecular weight distribution[27]. Notably, isoindigo-based low-bandgap polymers have been widely used in high-efficiency organic solar cells[28,29]. Similar to other SF polymers, IIDDT-Me contains a strong electron withdrawing group with significant contribution from a quinoidal resonance structure (Fig. 1b). This design strategy has been shown to reduce bond length alternation and yields low energy electronic states[30,31].

We have established that the IIDDT-Me polymer is a good candidate for SF, i.e., the energy conservation criteria for SF are likely to be met in this system, by calculating the electronic structure of a model ADADA (A = isoindigo and D = bithiophene) oligomer with density functional theory (DFT) and time-dependent DFT (TDDFT) (see Methods for details). We found that at the optimized ground state geometry, the lowest singlet transition ($S_1$) energy is 1.89 eV and the optimized lowest triplet state ($T_1$), calculated by unrestricted DFT, has energy of 0.95 eV above the ground state. These calculations suggest that this system is nearly isoenergetic for SF as dictated by the energy conservation requirement ($E_{S_1} \geq 2E_{T_1}$)[2]. As the energy of the exchange coupled triplet pair is expected to be less than twice the energy of the free triplet state calculated here[2,20,32,33], we expect SF to proceed with high yield in this system. Still, due to the near isoergicity, we expect the SF yield and kinetics to be highly sensitive to factors that affect the singlet energy (e.g., solvent polarizability and polymer aggregation).

The low triplet energy is consistent with several commonly used empirical design rules. For example, we have characterized the $S_1$ and $T_1$ transitions from TDDFT with the natural transition orbitals (NTOs) approach[34]. Using the NTO analysis, we find that the electron and the hole in both the singlet and triplet manifolds contain a high degree of spatial overlap (Fig. 1c). It has been suggested that this scenario will lead to a large singlet–triplet energy gap[35]. Furthermore, while the $S_1$ transition is delocalized over the whole oligomer, the triplet is primarily localized over the central isoindigo, with a small amount extending over to the neighboring thiophene. This result is similar to the previously reported PBTDO1 SF polymer[17], where the triplet is primarily localized on the thiophene dioxide acceptor unit. The spatial localization of the triplet suggests that the triplet prefers a different molecular geometry and hence it can be further stabilized with local structural distortion, similar to the polaron effects (Supplementary Fig. 1 and Supplementary Table 1). The stabilization energy is estimated to be 0.464 eV by comparing the triplet energy change (using unrestricted DFT) from the optimized ground state geometry to the optimized triplet state geometry. Importantly, the large stabilization energy suggests that dynamical structural relaxation will occur in the excited state of the polymer upon conversion of a singlet exciton into a triplet exciton.

The calculated $S_1$ energy matches the experimental absorption spectrum of IIDDT-Me (Fig. 1d) in good solvents, such as chloroform and dichlorobenzene (DCB). Using a second derivative analysis (Supplementary Fig. 2), we assign $S_1$ to the lowest energy absorption peak at 645 nm (calculated value =

660 nm) and observe a secondary vibronic peak at 595 nm near the net maximum[36]. However, we are able to significantly extend the low-energy absorption below 750 nm by aggregating the polymer in a worse solvent, such as trichloroethylene (TCE) or in thin film casts from the solution. The large redshift of the aggregate (>0.4 eV lowering of absorption maxima from DCB solution to film) absorption onset suggests that the energy conservation requirement for SF may not be met in the aggregate state, although aggregation will also modify the exchange interactions in the polymer.

**Triplet formation confirmed by time-resolved absorption spectroscopy.** We find that well-solubilized chains of IIDDT-Me efficiently produce triplet pairs via intramolecular SF in the solution (e.g., in DCB). Our analysis follows the generally accepted method for identifying iSF using a combination of transient absorption spectroscopy (TA), transient photoluminescence, and triplet sensitization measurements, which has been widely reported for small molecules and polymers[20,24,37]. The broadband TA data in DCB (Fig. 2a) displays an unusually rich dynamical evolution, from which we can identify the primary decay processes associated with the photoexcited singlet. The initial signal is dominated by ground state bleaching (GSB) of the $S_1$ state at 645 nm (Supplementary Fig. 2) and a broad stimulated emission signal (SE) with a minimum near 700 nm. Over time, the SE signal exhibits a large and continual redshift, and a new excited state absorption signal emerges on the ~10 ps timescale with a peak near 710 nm. However, the GSB minimum stays constant during the measurement, ruling out energy migration to lower energy polymer sites and indicating that electronic evolution of the excited states has occurred. Furthermore, the fast 10 ps decay kinetics that governs the rise of the excited-state absorption (ESA) are absent in GSB (Fig. 2b). The conservation of the GSB signal is in agreement with other investigations of intramolecular SF systems[37] and indicates that no loss of excited state population is occurring during this process, i.e., only the excited state electronic configuration is changing.

We deconvolute the overlapped spectral signatures of individual species (Fig. 2c), and model the population evolution with time (Fig. 2d) using global target analysis. Using a sequential decay model, we resolve three distinct time constants of 7.8, 58, and 190 ps, respectively, that describe the dynamical evolution of different electronic states of the polymer. The evolution-associated spectra, which represent the characteristic transient spectra associated with each time constant, are shown in Fig. 2c. While the initial and final states are highly distinct, the transition state resembles a combination of the two. Using these transient spectra with the determinations of the time-resolved photoluminescence and sensitization spectra (vide infra), we assign the state associated with the 7.8 ps time constant as the photoexcited singlet ($S_1$), the 190 ps state as the triplet pair (TT), and the intermediate 58 ps state ($S_1$ + TT) as representing a coexistence state, where both $S_1$ and TT are present in the ensemble. The coexistence of $S_1$ and TT states has been previously observed in molecular systems, such as tetracene, pentacene, and terrylenediimide derivatives[25,38,39]. The relative concentrations of $S_1$ and TT in the coexistence state are described by an equilibrium constant ($K$) that is determined by the forward ($k_{SF}$) and the reverse ($k_{TTA}$) rate constants describing the conversion between them:

$$S_1 \underset{k_{TTA}}{\overset{k_{SF}}{\rightleftarrows}} TT \quad K = \frac{k_{SF}}{k_{TTA}}$$

The equilibrium constant is not necessarily static in time and will be affected by anything that directly affects some combination of

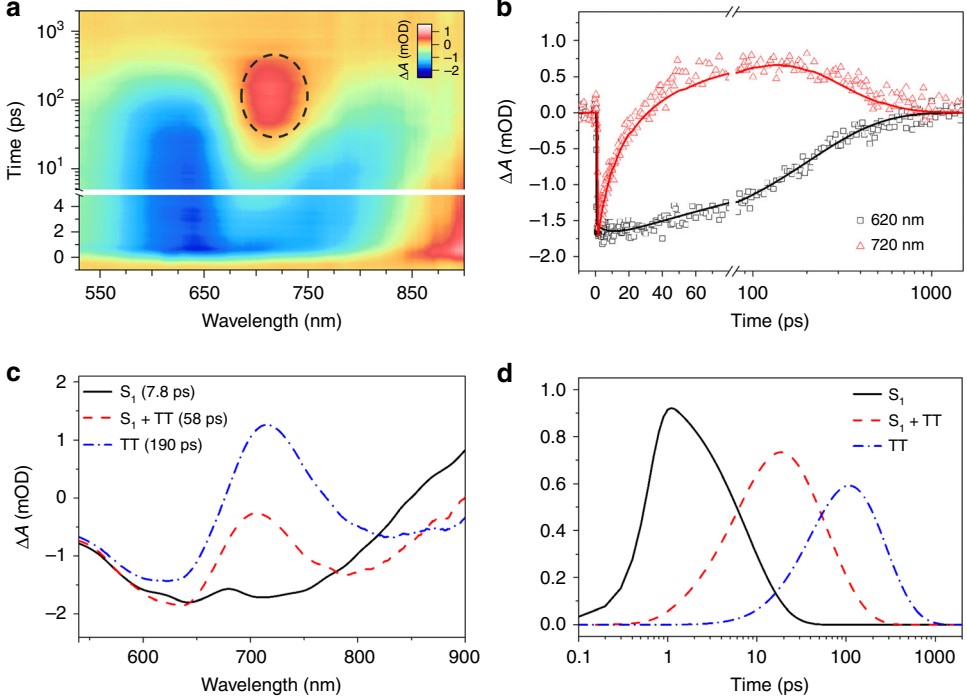

**Fig. 2** Transient absorption data. **a** Transient absorption of IIDDT-Me in the diluted DCB solution is shown in a pseudo-color plot, the dotted circle highlights the triplet transient signal. **b** The dynamics taken from GSB at 620 nm and ESA at 720 nm are shown with kinetic traces (open circles) and fits (solid lines). **c** A sequential global analysis model shows three species that we assign to the photoexcited singlet ($S_1$), triplet pair (TT), and singlet and triplet coexistence state ($S_1$ + TT). **d** The normalized population concentration versus time

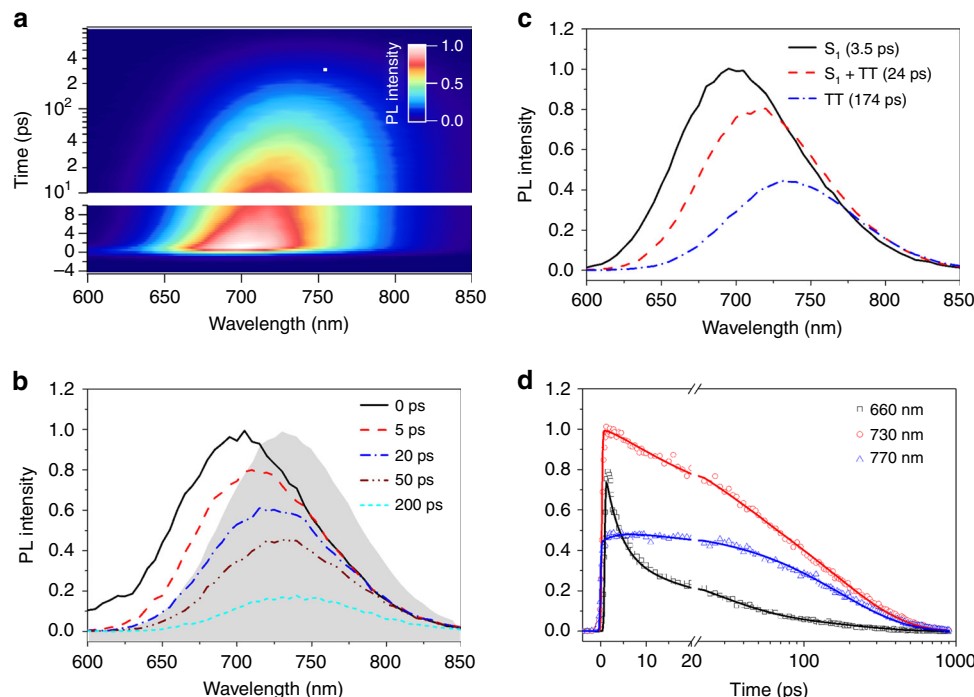

**Fig. 3** Transient fluorescence data. **a** Broadband transient photoluminescence of IIDDT-Me in dilute DCB solution. **b** The transient spectra extract at different time delays (solid lines) and the integrated PL spectra (filled gray curve). **c** Results of a three species sequential global analysis model showing the characteristic emission spectrum of singlet exciton ($S_1$), triplet pair (TT), and coexistence state ($S_1$ + TT). **d** Kinetic traces taken from the raw data set at 660, 730, and 770 nm (open circles) along with fits derived from global fitting (solid lines)

the energetic driving force or degree of charge transfer character. The fact that the final state in the overall dynamical evolution of the excited state has pure TT character implies that the energetics of SF is changing as a function of time.

**Delayed fluorescence and direct TT emission**. Our picture of dynamical evolution of the equilibrium constant describing the singlet and triplet populations is supported by transient photoluminescence measurements of IIDDT-Me, which indicate that

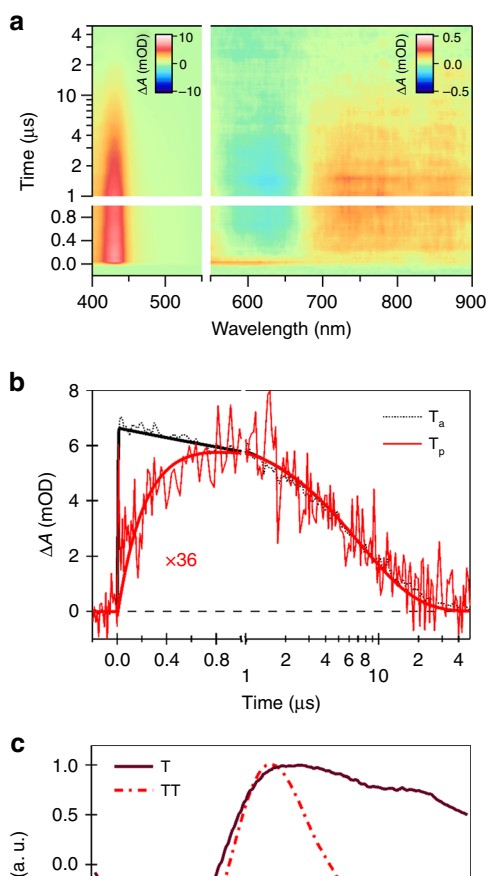

**Fig. 4** Triplet sensitization experiment. **a** Sensitized experiment of IIDDT-Me in the diluted DCB solution. **b** The dynamics taken at anthracene triplet ($T_a$) and polymer triplet ($T_p$) transient absorption region. **c** The lineshape of individual triplet and triplet pairs

the triplet pair is directly emissive and that its energy evolves in time. The net peak of the transient emission in dilute DCB solution (Fig. 3a and b) redshifts in time, similar to what we observed for the SE in transient absorption measurements. We apply a sequential global fitting scheme to the broadband transient photoluminescence, similar to what we have done for the transient absorption data. We find that the data are again well described by three distinct species, with time constants that closely match those extracted from TA (3.5, 24, and 174 ps). This data can be understood using the same scheme for conversion of singlet to triplet pairs as above, with the fast time constant representing the decay of the $S_1$ state with emission maximum at 700 nm, the longest time constant representing the TT state (735 nm), and a coexistence state ($S_1 + TT$) with an intermediate emission maximum. The dynamics associated with these states can be visualized by taking kinetic slices at different emission wavelengths. For example, the data at 660 nm primarily consist of the prompt emission associated with the decay of $S_1$, the data at 770 nm primarily reflect population dynamics of the triplet pair via delayed photoluminescence, and the data at 730 represents the shifting equilibrium constant.

These measurements are the first reports of direct TT emission in intramolecular SF materials, and confirm a recent theoretical

work suggesting that broken spatial symmetry can result in a highly emissive triplet pair that decays directly by photoluminescence[26]. As the symmetry of the molecule is reduced due to large differences in the donor and acceptor $sp^2$ carbon site energies, considerable configuration mixing occurs. This theory predicts that emission is strongest when the singlet state (1e–1h) is nearly in resonance with the TT state (2e–2h), similar to what is observed here. We note that a similar phenomenon has recently been reported in molecular crystals, where a distinct redshifted-delayed emission signal was observed and assigned to direct the TT emission[40,41]. While the prediction of direct TT emission in polymers is based on purely electronic considerations and does not involve phonon-mediated intensity borrowing arguments that have recently been invoked in molecular crystals[40], both reports suggest that symmetry breaking is a necessary condition for the observation of direct TT emission. In the scenario of an optically accessible TT state, the red-shifting emission peak directly corresponds to the evolution of the triplet pair from a higher average energy to a lower one. As the singlet energy remains constant as a function of time (Fig. 2), this means that we are directly measuring the change in the SF driving force ($\Delta E_{ST}$) as a function of time. We suggest that the change in driving force is primarily responsible for the dynamically increasing equilibrium constant, by enhancing the exciton fission rate constant (increasing exoergicity) and suppressing the exciton fusion one (increasing endoergicity).

**Comparing the triplets generated by iSF with native triplets.** We confirm our assignment of the triplet pair state using triplet sensitization measurements[42]. In addition to generating an assignment, this method can be used to identify the effects of Coulomb and exchange coupling on the triplet transient spectra and decay dynamics. Our evidence for triplet pair formation in the transient absorption spectra is supported by (1) the similarities between the triplet pair and free triplet spectra, (2) the more rapid decay of the triplet pair compared with the free triplet, (3) the presence of delayed photoluminescence and stimulated emission from the triplet pair, but not the free triplet, and (4) the disappearance of the triplet pair signal in the aggregated state. We will address each of these points sequentially.

Using anthracene as a triplet sensitizer, we obtain the free triplet ($T_1$) spectra using nanosecond transient absorption measurements (Fig. 4a and Supplementary Note 1). The collisional energy transfer dynamics to the polymer is monitored via the anthracene triplet excited state absorption band at 420 nm and the polymer triplet absorption band in the NIR. The conversion dynamics here are characteristic of the kinetic limit where the polymer triplet lifetime is much shorter than the sensitization (energy transfer) time constant. In this regime (see SI for more details on the solution of the differential equations in this limit), the rise of the polymer triplet signal ($T_p$, 0.23 μs) reflects the native triplet lifetime, and the decay matches that of the sensitizer ($T_a$, 7.27 μs, Fig. 4b). A second important consequence of the short triplet lifetime in IIDDT-Me is that the instantaneous population [$T_p$] of polymer triplets is low, achieving a maximum of ~3% at 800 ns if 100% triplet transfer is assumed. In other words, triplets rapidly decay to the ground state via the intersystem crossing (ISC) soon after transfer to the polymer.

The absorption feature of the triplet pairs generated by iSF was found to be comparable to the native triplet signal obtained through sensitization. As triplet pairs generated by iSF systems are largely in the strong exchange coupled limit[43], we expect to observe non-neglible differences between TT and $T_1$. Indeed, the lifetime of native single-triplet state (0.23 μs) of IIDDT-Me is

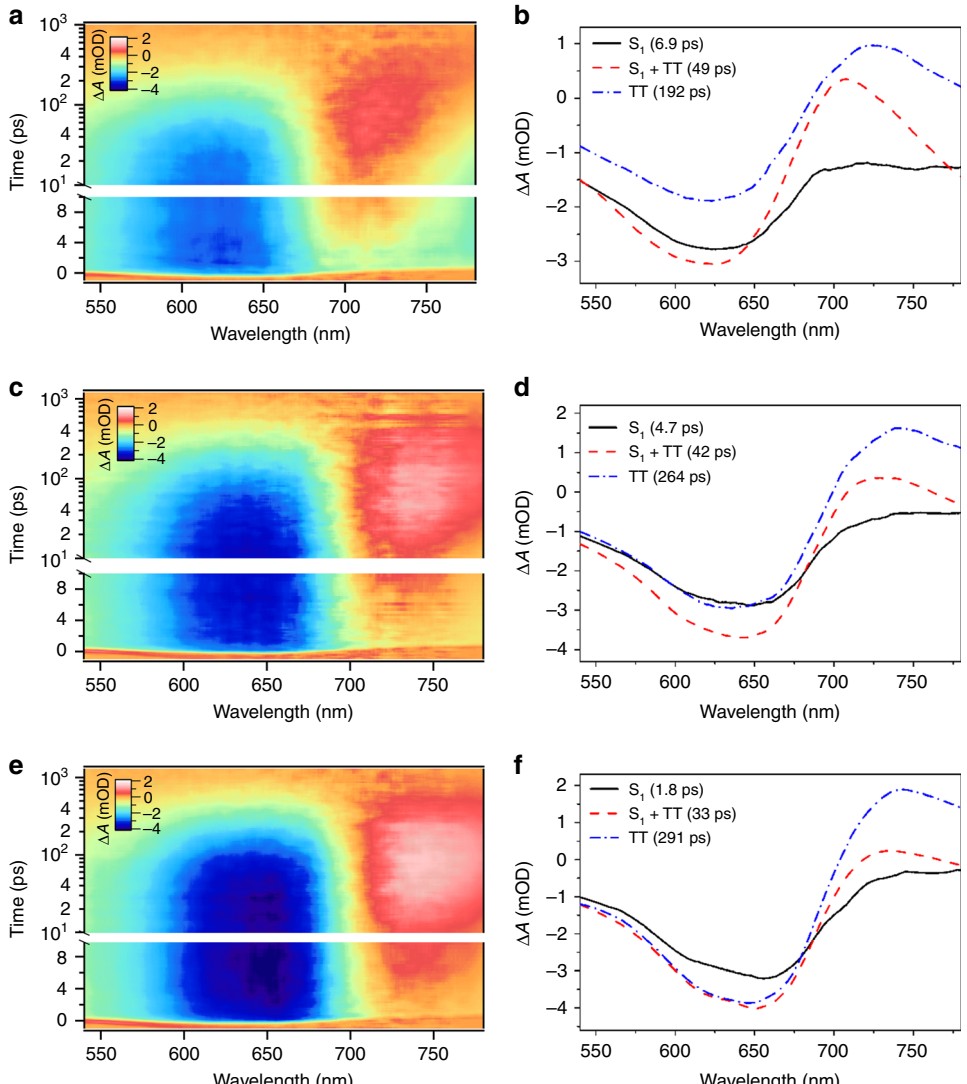

**Fig. 5** Low temperature transient absorption. Transient absorption of IIDDT-Me in the CHCl$_3$ solution at room temperature (**a**), 270 K (**c**) and 255 K (**e**) are shown in pseudo-color plots, and the corresponding global analysis results are shown in (**b**), (**d**), and (**f**), respectively

~100× longer than the lifetime of triplet state generated by iSF (~200 ps, Supplementary Note 1 and Supplementary Fig. 3) in general agreement with all other measurements of intramolecular SF systems[13,20,24,25]. This rules out the fast formation of free triplets via ISC or other kinetic processes[44]. The difference between the free triplet and triplet pair spectra (Fig. 4c) primarily results from the presence of SE in the TA data. This is strong evidence supporting recent claims of a directly optically accessible TT state. The delayed PL signal is a strong evidence for the isoergicity of the singlet and triplet pair. However, S$_1$ and T$_1$ conversion via another process (e.g., thermally activated delayed fluorescence) is ruled out by the low T$_1$ energy.

Due to the coexistence and shifting equilibrium between the singlet and the triplet pair, it is difficult to articulate a triplet pair yield in the traditional sense. We observe no parasitic process that would compete with the SF, and the rate constant for decay of the photoexcited singlet (~7 ps) occurs before any material loss of the excited state population (monitored using the GSB decay dynamics). This method has been validated for other SF systems[10–14]. As such, it is probable that all photoexcited singlets are converted to the triplet pair state during their lifetime. Still, the final state consisting of majority triplet pairs evolves with a time constant of ~50 ps, during which ~20% of the excited state

population has returned to the ground state via prompt and delayed photoluminescence. Using the integrated triplet pair population in the final TT-only state as a lower bound for the effective triplet pair yield gives a value of 80% triplet pairs. The lifetime of the TT state is too short to perform time-resolved electron spin resonance studies that are uniquely capable of identifying the total spin angular momentum. As the triplets are not able to separate over long distances in this constrained system, no obvious long-lived free triplets were detected in the solution. However, the harvestable triplet yield could be as high as 160–200% in an appropriately engineered photovoltaic device.

**The SF rate depends on temperature.** Unlike most reported SF systems[38,40,45,46], IIDDT-Me exhibits a temperature-dependent rate constant for SF. In addition, we have found the SF rate constants to be weakly solvent dependent. To show both of these effects, transient absorption studies in dilute chloroform solution were conducted at various temperatures (Fig. 5). The overall carrier dynamics of IIDDT-Me in chloroform are similar to what is observed in DCB, with a slightly shorted time constant for SF and the coexistence state (~7 and ~50 ps), but a nearly identical time constant for triplet pair decay (~190 ps). As temperature is

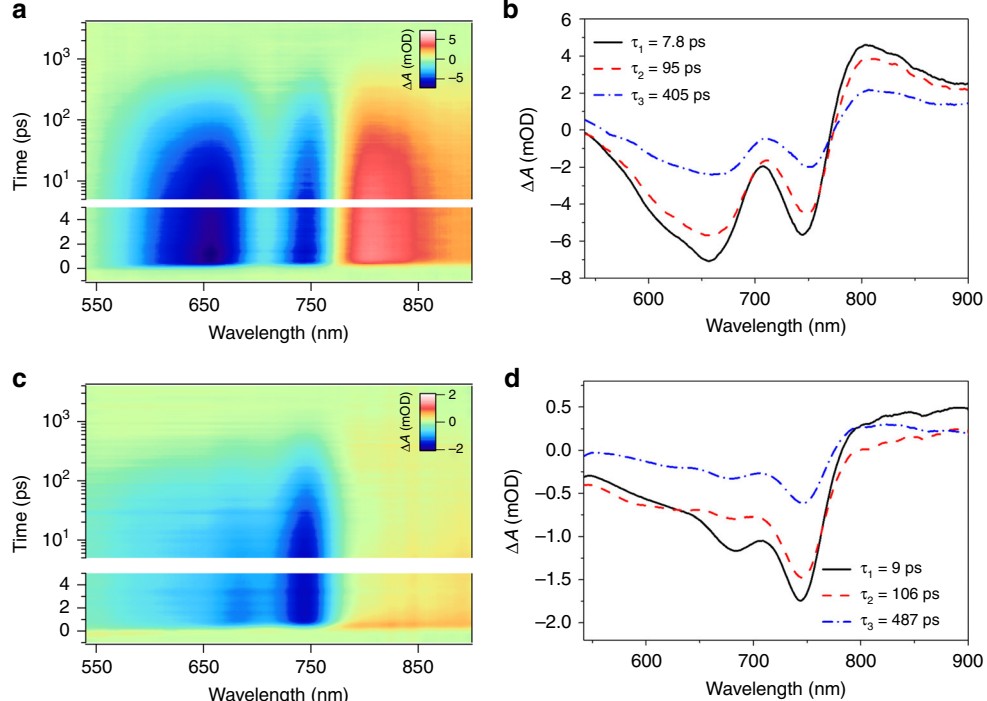

**Fig. 6** Transient absorption spectroscopy in aggregate states. **a** Transient absorption of IIDDT-Me in the DCB film is shown in a pseudo-color plot. **b** Global analysis of IIDDT-Me in the DCB film. **c** Transient absorption of IIDDT-Me in the TCE solution is shown in a pseudo-color plot. **d** Global analysis of IIDDT-Me in the diluted TCE solution

decreased (Fig. 5 and Supplementary Fig. 4), we observe a decrease in both the singlet lifetime (from ~7 ps at room temperature to ~2 ps at 250 K) and the lifetime of the coexistence state (~50 ps at room temperature vs. ~30 ps at 250 K). Both these changes are consistent with a slightly more exothermic driving force for SF at lower temperature. The dependence of the SF rate constant with temperature is unusual, different from other studies that show that SF is largely independent of temperature[38,45,46]. This implies that an additional consideration is necessary in these systems to describe the overall SF mechanism.

We suggest that the observed trend of increasing SF rate with decreased temperature is derived from slight conformational changes in the molecule that modify the energetic driving force and/or charge transfer character that affect the SF rate constants. Changes in the ground state equilibrium geometry (a function of solvent and temperature) can be described as static effects that modify the driving force for SF rate. Similarly, relaxation of the polymer conformation in the excited state leads to dynamic changes that modify the relative populations of the singlet and triplet state as a function of time in the coexistence state. We note that our DFT calculations, which suggest large reorganization energy between the singlet and triplet manifolds, provide a plausible explanation that is consistent with our description of dynamical evolution of the polymer conformation in the excited state. Collectively, these data indicate a non-trivial sensitivity of the electronic structure to small conformation changes in the polymer.

**Aggregation quenched SF.** The aforementioned studies were conducted under high dilution condition, and UV–Vis absorption at elevated temperature verified that the polymer chains behave independently at this concentration (Supplementary Note 2 and Supplementary Fig. 5). However, SF in IIDDT-Me is quenched due to aggregation in both solutions and thin films. As noted above, aggregation is accompanied by a large redshift of the singlet

energy (Fig. 1d). The magnitude of this redshift (~400 meV) is too large to be accounted for by screening (dielectric) effects[47] and much larger than those observed for a related class of polymers[27]. Similar to the effects of temperature and solvent, this suggests that modification of the equilibrium geometry of the polymer is occurring upon aggregation. As an example, we show the transient absorption data for TCE solution (a poor solvent for IIDDT-Me) and a thin film cast from DCB (raw data in Fig. 6a, c and the global fitting results in Fig. 6b, d). Unlike the single polymer chain data, the aggregate excited states decay via a conventional process for conjugated polymers, with a dynamical evolution of GSB to lower the energy sites and an overall longer excited state lifetime. Similar behaviors were also observed for the transient absorption data of IIDDT-Me films casted from $CHCl_3$ and TCE (Supplementary Fig. 6). We are not able to identify any emergence of the triplet excited state absorption in these systems.

From the linear and transient optical data, we conclude that the energetic criteria for SF are no longer satisfied in the aggregate state. This implies that changes in the exchange energy are small compared with the changes in the singlet energy. Similar concerns about the effects of strong chromophore coupling on the energy and the character of the singlet state has been expressed in small molecule systems[5,48]. Besides helping to assign the triplet pair, this result has significant consequences for the design of SF devices. For intramolecular SF materials that are approximately isoenergetic, appropriate triplet harvesting schemes must be employed to prevent strong aggregation and the loss of SF. This appears to be especially true for polymeric systems, which exhibit an enhanced sensitivity of the SF process to environmental factors.

## Discussion

Polymer inter-chain interactions must be carefully engineered to achieve efficient intramolecular SF and this can be described as a new rule for designing low-bandgap conjugated polymers capable

of efficient iSF. The development of this design rule means that many existing D–A polymers can be re-examined in the context of SF. While most OPV materials have been designed to promote strong intermolecular packing to achieve better charger transfer, our results suggest that an opposite approach is needed for efficient SF materials. We suggest that small side chain modifications can be used to suppress intermolecular packing and bias the formation of triplet pairs through iSF for the large library of reported D–A polymers.

## Methods

**Materials.** IIDDT-Me ($M_n = 20.1$ kDa, $Đ = 3.9$) was synthesized as described previously[27].

**Transient absorption spectroscopy.** TA was performed using a commercial femtosecond pump–probe system (Transient Absorption Spectrometer, Newport Corporation). Laser pulses at 1040 nm with <400 fs duration were generated by a 200 kHz amplified laser system (Spirit 1040-8-SHG, Newport Corporation). The probe beam was a white light continuum beam spanning 500–950 nm spectral region, created by focusing a fraction of the 1040 nm fundamental output onto a YAG crystal. The rest of the output generated the pump pulses at 520 nm by second harmonic generation. The pump–probe delay was controlled by a mechanical delay stage. The excitation fluence in each measurement was ~75 μJ cm$^{-2}$. To confirm that the dynamics is independent of the pump intensity, the fluence dependent experiment was conducted and the results are shown in Supplementary Note 3 and Supplementary Fig. 7. The species determination for global analysis is detailed in Supplementary Note 4 and Supplementary Figs. 8, 9 and 10.

**Triplet photosensitization.** Triplets were generated by excitation of an excess of anthracene (360 nm), which undergoes ISC, and were subsequently transferred via diffusional collisions to the polymer IIDDT-Me. In this way, a single triplet could be transferred to the polymer, in direct contrast to optical excitation, which produced a triplet pair in the case of SF materials. Then, the solution was optically probed to reveal the induced absorption spectrum of the triplet and the native triplet lifetime. For these measurements, a 1 kHz Ti:Sapphire-based amplified system was used along with a commercial nanosecond transient absorption spectrometer (EOS, Ultrafast Systems).

**Ultrafast photoluminescence spectroscopy.** Ultrafast photoluminescence decay kinetics was measured by the upconversion technique. Briefly, a dilute DCB solution was resonantly excited with a 520 nm, <400 fs laser pulse. The spontaneous emission was collected by use of a 570 nm long pass filter and mixed with a 1040 nm, <400 fs pulse in a non-linear crystal in a geometry optimized for sum frequency generation. A 350–450 nm filter was used to remove the 1040 nm pulse, thus photoluminescence after 800 nm was cut. The magnitude of the upconverted optical signal was proportional to the instantaneous photoluminescence intensity and was detected as a function of delay between the excitation and the 1040 nm pulses. The delay was controlled by a mechanical delay stage. The detailed global analysis results were shown in Supplementary Figs. 11 and 12.

**Computation.** The ground state geometry is optimized using DFT, and the excited states are calculated with linear response time-dependent DFT (TDDFT) at the optimized ground state geometry. The relaxed lowest triplet state is further optimized with a spin-polarized DFT calculation. All calculations are performed with the Gaussian 16 package (Rev. A.03) using the hybrid B3LYP functional and the 6-311G* basis set. A solvent (chlorobenzene) reaction field simulated by the default polarizable continuum model (PCM) is also employed[49]. We then use the NTO approach to characterize the nature of the lowest singlet and triplet states.

**Data availability.** The authors make a statement that the data presented in this article are available from the corresponding authors on reasonable requests.

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

## Acknowledgements

J.X. acknowledges the financial support from the National Natural Science Foundation of China (NSFC, 21502147, 51773160) and the generous start-up funds from Wuhan University of Technology (No. 40122004). This research used resources of the Center for Functional Nanomaterials, which is a U.S. DOE Office of Science Facility, at Brookhaven National Laboratory under Contract No. DE-SC0012704.

## Author contributions

J.X. and M.S. conceived and designed this project. J.H. measured the majority of the femtosecond TA and PL data presented in the manuscript, and M.S. performed triplet sensitization and low temperature measurements. K.X., L.S., and G.H. assisted with data collection and analysis and performed TA measurements in CHCl₃solution. Q.W. performed the calculations. J.Y.W. and J.P. synthesized the polymers for this project. J.H., Q. W., J.X., and M.S. analyzed the data and co-wrote the manuscript.

## Additional information

**Competing interests:** The authors declare no competing interests.

