## [Peer Review File · Nature Communications]

Reviewers' comments:

Reviewer #1 (Remarks to the Author):

The authors describe a new IIDDT-ME polymer for which they observe singlet fission in good solvents, but when aggregated by a bad solvent or in a thin film, they find that singlet fission is quenched due to a lower singlet exciton energy. They draw conclusions about the role of molecular interactions for singlet fission. The authors support their claims with transient absorption, time-resolved PL and triplet quenching experiments.

I find the results convincing, and the conclusions about the design rules for i-SF materials relevant for the field. The works should hence be published in Nature Communications, but not before my main criticism below is addressed.

My main criticism concerns the assignment of the spectral features in both the TA and TR-PL of the "good solvent" data. The authors assign three features in both cases, one for the S1 state, one for the TT state, and one mixed state. In both cases, however, the spectral features of the mixed state are clearly resembling a linear combination of the S1 and the TT state. Hence, there are only two states apparent in the spectra, and the mixed state is overfitting the data.

This assignment of only two features would also resolve an apparent problem with the data. The authors write that the red shift they observe when K , the equilibrium constant, changes, is due to a change in the triplet pair energy. However, I cannot see how the triplet pair energy would change, certainly not on the scale of many picoseconds. Instead, their data suggests that there are only two states of well-defined energy, and the apparent red-shift is simply a transition from the higher-energy state (S1) to the lower-energy (TT) state.

Apart from this main comment, a few smaller comments:

- 1) The authors put a lot of weight on the DFT calculations, and they quote the excited-state energies with great precision (single meV). I do not think that DFT can predict excited states that accurately.
- 2) In the aggregated state the Si energy decreases, and the authors directly relate that to a reduced driving energy for singlet fission. However, the exchange energy is probably also changing with aggregation, and so is the difference in energy between the S1 and T1 state, thus, while it is likely that the driving energy reduces, it is nowhere near certain.
- 3) ESA is used without definition
- 4) The yield calculations are only done for TT. The far more interesting yield is the one of free triplets (ie T1+T1). The authors should make that difference very clear and speculate on the T1 yield.

Reviewer #2 (Remarks to the Author):

The manuscript reports a quenching of intramolecular singlet fission in isoindigo-based donor-acceptor polymers upon aggregation.

This main result appears novel, but not of extreme importance and multidisciplinary relevance/interest typical for Nature Communications publications.

Therefore, I recommend to submit the manuscript to a more appropriate journal and refrain from a detailed review.

Reviewer #3 (Remarks to the Author):

The manuscript entitled "New insights into the design of conjugated polymers for intramolecular singlet fission" by Xia and co-workers represents an interesting study of polymers that are allegedly

capable of intramolecular singlet fission.

The paper describes TA and TRPL data which is globally analysed. Though the concentration was not clearly stated, we are led to believe that the polymer chains behave independently. Could this be proven?

The biggest problem with this manuscript is the interpretation of the spectroscopic data. For example, the TA data is decomposed into three signatures: S1, TT and "S1+TT", the latter denoted a "co-existence state". What is this coexistence state? If the ensemble contains both S1 and TT, as it will at intermediate times, then those spectra will account for the data.

What is clear is that the photoluminescence shifts to the red with time, and this is mirrored in the SE signal. I find it more likely that this represents slow structural evolution of the polymer chain in the S1 state. This could represent adiabatic passage into a TT state, as in Tayebjee's paper 10.1039/C3CP52609G.

Overall it does not seem that the claims in the manuscript are supported by the data and its interpretation, leading to doubt into any "insight" gained. That the triplet state energy changes in time seems highly doubtful. Rather, it appears that the excited state evolves to one of lower energy and might eventually have TT character.

Low temperature experiments in glassy matrices and magnetic field dependences would be of interest to clear up some of these matters.

Further concerns are detailed below roughly in the order that they appear.

It is stated that reference 16 is the "first reported iSF system." But, see J Am Chem Soc. 2015 Jul 22; 137(28): 8965-72. doi: 10.1021/jacs.5b04986. Epub 2015 Jul 14.

Quantitative Intramolecular Singlet Fission in Bipentacenes. This was published many months before reference 16, and has one overlapping author with the current work!

The mention of kinetic competition with excimer dissociation would be a good place to cite the recent Nature Chemistry paper from Dover et al. doi:10.1038/nchem.2926

Why would spatial localization of the triplet suggest that it can be further stabilized? Were the calculations of the stabilization energy done at the same level of theory? Both unrestricted?

"*vide infra*" should be written "*vide infra*" and is not an adjective. It is an order: "see below".

Why does not the GSB increase as fission proceeds? Surely a triplet takes away two ground states? Or is the TT not really a separated triplet?

I would soften the language regarding the DFT result, and clearly state the function and basis set used. DFT will tell if fission is plausible, but is not accurate enough to say much else.

The authors claim that 20% of the excited state population radiatively decays within 50ps. This implies a radiative lifetime of 250ps,

160% triplet pair yield would imply 320% triplet yield. I guess the authors mean 160% triplet yield, and therefore 80% triplet pair yield.

There are a few typos that I have spotted:

"isoindio" in the abstract.

Response to the reviewers:

Reviewer #1:

My main criticism concerns the assignment of the spectral features in both the TA and TR-PL of the “good solvent” data. The authors assign three features in both cases, on for the S₁ state, one for the TT state, and one mixed state. In both cases, however, the spectral features of the mixed state are clearly resembling a linear combination of the S₁ and the TT state. Hence, there are only two states apparent in the spectra, and the mixed state is overfitting the data.

Response: Our analysis method is able to identify the “characteristic spectra” associated with the primary time constants present in the data set. In other words, the number of time constants determines the number of mathematically linearly independent species, not the actual number of physically distinct species present in the data (e.g., singlet and triplet pair). It is possible to have the same species (e.g., triplet pair) exhibit two or more different time constants, in which case two of the mathematically distinct species will have the same “characteristic spectra”. For an example, see our work on singlet fission dimers (doi://10.1021/jacs.5b04986).

In this case, we have three well separated time constants that after global fitting give the “characteristic spectra” shown in the manuscript. The mixed singlet/triplet pair state is assigned based on the fact that its “characteristic spectra” is a linear combination of the singlet and triplet pair “characteristic spectra” together with our analysis of the transient photoluminescence data that shows two distinct emitters.

To further show that the data are fit appropriately, we present here for review only an equivalent analysis with only two constants (Figure R1). The two time-constant fits do not satisfactorily describe the intermediate dynamics observed (Figure R1d) and the full residual (Figure R1f) clearly shows that not all features are accounted for.

Figure R1. (a) Transient absorption of IIDDT-Me in the diluted DCB solution is shown in a pseudo-color plot. (b) The global fitting of three exponentials. (c) The global fitting of two exponentials. (d) The dynamics taken from 700 nm (gray circles) and 810 nm (purple square), and the kinetic traces are shown in two time-constant fits (black line at 700 nm and blue line at 810nm) and three time-constant fits (gray line at 700 nm and purple line at 810 nm), respectively. (e) The residual transient absorption

after global fitting of three exponentials. (f) The residual transient absorption after global fitting of two exponentials.

We have edited the manuscript to clarify our description of the analysis. For example, we have edited the description of the analysis to read:

“Using a sequential decay model, we resolve three distinct time constants of 7.8 ps, 58 ps and 190 ps respectively that describe the dynamical evolution of the different electronic states of the polymer. The evolution associated spectra, which represent the characteristic transient spectra associated with each time constant, are shown in Fig. 2c.”

This assignment of only two features would also resolve an apparent problem with the data. The authors write that the red shift they observe when K , the equilibrium constant, changes, is due to a change in the triplet pair energy. However, I cannot see how the triplet pair energy would change, certainly not on the scale of many picoseconds. Instead, their data suggests that there are only two states of well-defined energy, and the apparent red-shift is simply a transition from the higher-energy state (S1) to the lower-energy (TT) state.

Response: We believe that slow structural relaxation occurs on the 10s of picosecond time scales that affects the relative energy of the singlet and triplet pair as a function of time, changing the driving force for singlet fission and shifting the equilibrium constant over time to favor a pure TT state. If direct conversion of a pure singlet state to a pure triplet pair state were occurring, the intermediate state wouldn't exist and the decay of the singlet would exactly match the rise of the triplet. Please see also doi://10.1021/jacs.5b04986, in this scenario, two time constants would be sufficient to describe the full data set. This is not the case here. Rather, the behavior here reflects an ensemble with a distribution of singlets and triplets whose relative concentration is given by the equilibrium constant.

In addition to an extensive new discussion based on temperature dependent measurements, we have added the following discussion:

“The equilibrium constant is not necessarily static in time and will be affected by anything that directly affects some combination of the energetic driving force or degree of charge transfer character. The fact that the final state in the overall dynamical evolution of the excited state has pure TT character implies that the energetics of singlet fission is changing as a function of time.”

Apart from this main comment, a few smaller comments:

1) The authors put a lot of weight on the DFT calculations, and they quote the excited-state energies with great precision (single meV). I do not think that DFT can predict excited states that accurately.

Response: We have modified the DFT discussion section in the revision to clarify that the purpose of the DFT calculations are simply show that they are consistent with the idea that this material meets the energy conservation criterion for singlet fission as the transient data implies. It is one of many pieces of evidence used to support our data interpretation that the singlet and triplet pair are nearly isoergic. We have reduced the precision of the numbers reported.

2) In the aggregated state the Si energy decreases, and the authors directly relate that to a reduced driving energy for singlet fission. However, the exchange energy is probably also changing with aggregation, and so is the difference in energy between the S1 and T1 state, thus, while it is likely that the driving energy reduces, it is nowhere near certain.

Response: The reviewer is correct in that exchange interactions are also affected by aggregation. Our explanation here is based on the empirical observation that the dramatic shifts in the singlet energy (> 400 meV) are accompanied by a suppression of the singlet fission decay channel. This necessarily indicates that the change in the triplet energy is less than half the change in energy of the singlet. We have modified the discussion to note that aggregation affects the exchange energy as well, but that the data indicated that the net red shift of the triplet energy is small compared to the change in the singlet.

We have modified the discussion in the manuscript to read, “The large redshift of the aggregate (> 0.4 eV lowering of absorption maxima from DCB solution to film) absorption onset suggests that the energy conservation requirement for singlet fission may not be met in the aggregate state, although aggregation will also modify exchange interactions in the polymer.”

In discussing aggregates, we write:

“From the linear and transient optical data, we conclude that the energetic criteria for singlet fission are no longer satisfied in the aggregate state. This implies that changes in the exchange energy are small compared to the changes in the singlet energy.”

3) ESA is used without definition

Response: We have now defined ESA (excited-state absorption) in the manuscript.

4) The yield calculations are only done for TT. The far more interesting yield is the one of free triplets (ie T1+T1). The authors should make that difference very clear and speculate on the T1 yield.

Response: We have added the following discussion to the manuscript:

“The lifetime of the TT state is too short to perform time resolved electron spin resonance studies that are uniquely capable of identifying the total spin angular momentum. As the triplets are not able to separate over long distances in this constrained system, no obvious long lived free triplets were detected in solution. However, the potential yield could be as high as 160% in an appropriately engineered photovoltaic device.”

Reviewer #3:

The manuscript entitled "New insights into the design of conjugated polymers for intramolecular singlet fission" by Xia and co-workers represents an interesting study of polymers that are allegedly capable of intramolecular singlet fission.

The paper describes TA and TRPL data which is globally analysed. Though the concentration was not clearly stated, we are led to believe that the polymer chains behave independently. Could this be proven?

Response: For the data presented in the manuscript, the concentration of solution is about 50 μM . The dynamics are independent of concentration over the measurable range of 10 - 100 μM . To further prove that there is no aggregation even at these low concentrations, we monitored the UV-Vis absorption at elevated temperature and verified that no spectral changes occurred.

We have added the following discussion to the manuscript:

“The aforementioned studies were conducted under high dilution condition, and UV-Vis absorption at elevated temperature verified that the polymer chains behave independently at this concentration (Fig. S5).”

And we have added the figure to the SI (Figure S5):

Figure S5. UV-Vis absorption of IIDDT-Me in dichlorobenzene at elevated temperature.

The biggest problem with this manuscript is the interpretation of the spectroscopic data. For example, the TA data is decomposed into three signatures: S1, TT and "S1+TT", the latter denoted a "co-existence state". What is this coexistence state? If the ensemble contains both S1 and TT, as it will at intermediate times, then those spectra will account for the data.

Response: Yes, the “co-existence state” contains both the S_1 and TT states and interconversion between the two is occurring during the intermediate time frame. As the reviewer suggests, this means at any instantaneous time (snapshot) taken between 10 – 50 ps, a distribution of S_1 and TT exist in the ensemble with their relative concentrations determined by the equilibrium constant for the forward and back reactions. As the reviewer suggests, the transient spectra of the intermediate species accounts for the existence of both the S_1 and TT states. In fact, this is one of the primary pieces of evidence for the mechanism described in the manuscript.

We have modified the discussion to clarify these points. For example:
“... we assign the state associated with the 7.8 ps time constant as the photoexcited singlet (S_1), the 190 ps state as the triplet pair (TT), and the intermediate 58 ps state ($S_1 + TT$) as representing a coexistence where both S_1 and TT are present in the ensemble. The relative concentrations of S_1 and TT in the coexistence state are described by an equilibrium constant (K) that is determined by the forward (k_{SF}) and reverse (k_{TTA}) rate constants describing the conversion between them:”

We note that an important but subtle distinction must be made between the situation described above and the situation where an inhomogeneous ensemble yields a distribution of time constants. Our polymer exists as identical entities in the form of single isolated chains with a relatively narrow molecular weight distribution. As such, the equilibrium picture is the correct way to interpret this data.

What is clear is that the photoluminescence shifts to the red with time, and this is mirrored in the SE signal. I find it more likely that this represents slow structural evolution of the polymer chain in the S_1 state. This could represent adiabatic passage into a TT state, as in Tayebjee's paper 10.1039/C3CP52609G.

Overall it does not seem that the claims in the manuscript are supported by the data and its interpretation, leading to doubt into any "insight" gained. That the triplet state energy changes in time seems highly doubtful. Rather, it appears that the excited state evolves to one of lower energy and might eventually have TT character.

Low temperature experiments in glassy matrices and magnetic field dependences would be of interest to clear up some of these matters.

Response: We believe that our interpretation is largely consistent with the comments of the reviewer, though it seems we have not communicated our message clearly enough. As the reviewer suggests, we do believe that slow structural relaxation occurs on the 10s of picosecond time scales. This relaxation is suggested by our DFT calculations, which predict a large stabilization energy of the triplet upon structural relaxation. However, this relaxation primarily affects the relative energy of the singlet and triplet pair as a function of time, changing the driving force for singlet fission and shifting the equilibrium constant over time to favor a pure TT state.

As suggested by the reviewer, we have performed additional low temperature transient absorption measurements that rule out energy migration of the singlet state alone. We observe an increase in the formation of the low energy triplet pair as temperature is decreased. The decrease in both the singlet lifetime (from ~ 7 ps at room temperature in chloroform to ~ 2 ps at 250 K) and the lifetime of the

coexistence state (~ 50 ps at room temperature vs. ~ 30 ps at 250 K) is consistent with a slightly more exothermic driving force at lower temperature (due to slight conformational changes or slight changes in the dielectric properties of the solvent). It is inconsistent with migration of singlet excitons to lower energy sites via a diffusive type process, which is an activated process and would slow down at room temperature. These data have been added to the manuscript and SI and are now discussed in detail the main manuscript in the sections entitled, “**The singlet fission rate depends on temperature**” and “**Aggregation quenched singlet fission.**”

Collectively our data reinforce the idea that structural and energy relaxation will affect the driving force for singlet fission, and that we can separately identify *static* effects (changes in the ground state equilibrium geometry) and *dynamic* effects (evolution of the excited state geometry with time). The evidence can be summarized in the following way. **1)** There is a large change in the energy and peak width of the linear optical absorption upon aggregation. The magnitude (~ 400 meV) is too large to be accounted for by screening (dielectric) effects and as such, suggests a modification of the equilibrium geometry (static effect). One consequence of this change in the electronic structure is the disappearance of singlet fission. **2)** The dependence of the singlet fission rate constant with temperature is unusual, different from the paper by Tayebjee et al. (cited in the revision) as well as other studies that show that singlet fission is largely independent of temperature (doi://10.1038/nchem.2856, doi://10.1038/nphys3909). This implies that an additional consideration exists in these systems, which we assign to small modifications of the polymer conformation (static effects). **3)** By correlating the transient absorption and emission signals, we are able to identify the formation of a triplet-singlet equilibrium followed by relaxation into a pure triplet pair state, consisting with a changing equilibrium constant. The change in the equilibrium constant requires a change in the forward singlet fission rate constant, which low temperature measurements indicate is extremely sensitive to geometry (dynamic effects).

We have added to and clarified the discussion in the manuscript around these points, and emphasize the unifying principles between our different observations for the modification of singlet fission as a function of solvent, aggregation state, and temperature. In addition, the following figures have been added into the revised manuscript (Fig. 5) and supporting information (Fig. S4):

Figure 5 Low temperature transient absorption. Transient absorption of IIDDT-Me in the CHCl_3 solution at room temperature (a), 270 K (c) and 255 K (e) are shown in pseudo-colour plots, and the corresponding global analysis results are shown in (b), (d) and (f), respectively.

Figure S4 Temperature dependent kinetics of IIDDT-Me in the CHCl_3 solution: (a) room temperature; (b) 270 K; (c) 255 K.

We note that magnetic field effects cannot be used in *intramolecular* singlet fission systems with donor and acceptor type of structure to modify the rate constants because of the strong exchange coupling present in these systems (<https://doi.org/10.1103/PhysRevB.94.045204>). External magnetic fields can only be used to change experimental observables (photoluminescence, photocurrent, etc.) in weak exchange coupled systems (such as crystals) where the singlet and quintet state mix due to spin dipolar interactions.

It is stated that reference 16 is the "first reported iSF system." But, see J Am Chem Soc. 2015 Jul 22;137(28):8965-72. doi: 10.1021/jacs.5b04986. Epub 2015 Jul 14. Quantitative Intramolecular Singlet Fission in Bipentacenes. This was published many months before reference 16, and has one overlapping author with the current work!

Response: We have simply inserted the wrong pointer to a reference. The correct reference is Busby et. al, Nature Materials 14, 426–433 (2015) (Ref. 17). This preceded the JACS work referenced above, we have revised this in the revision.

The mention of kinetic competition with excimer dissociation would be a good place to cite the recent Nature Chemistry paper from Dover et al. doi:10.1038/nchem.2926

Response: We have added the Nature Chemistry paper by Dover et. al in the revision (Ref. 28).

Why would spatial localization of the triplet suggest that it can be further stabilized? Were the calculations of the stabilization energy done at the same level of theory? Both unrestricted?

Response: The spatial localization of the triplet suggests that the triplet prefers a different molecular geometry than the singlet and hence it can be further stabilized with local structural distortion, similar to polaron effects. We clarified this in the main manuscript.

The calculations of stabilization energy are done at the same level of theory using unrestricted DFT.

"vida infra" should be written "vide infra" and is not an adjective. It is an order: "see below".

Response: We have changed it in the revision.

Why does not the GSB increase as fission proceeds? Surely a triplet takes away two ground states? Or is the TT not really a separated triplet?

Response: In intramolecular singlet fission systems, the GSB is conserved during singlet fission. This results from electronic correlations in both the singlet and triplet manifolds. In a simple picture, both the singlet and triplet pair extend across multiple

chromophores (repeat units). We have addressed this issue in detail in a previous manuscript: doi://10.1021/jacs.5b04986.

We have modified the manuscript to read, “The conservation of the GSB signal is in agreement with other investigations of intramolecular singlet fission systems and indicates that no loss of excited state population is occurring during this process, i.e., only the excited state electronic configuration is changing.”

I would soften the language regarding the DFT result, and clearly state the function and basis set used. DFT will tell if fission is plausible, but is not accurate enough to say much else.

Response: We have modified the DFT discussion section in the revision to clarify that the purpose of the DFT calculations are simply show that they are consistent with the idea that this material meets the energy conservation criterion for singlet fission as the transient data implies. It is one of many pieces of evidence used to support our data interpretation that the singlet and triplet pair are nearly isoergic.

The authors claim that 20% of the excited state population radiatively decays within 50ps. This implies a radiative lifetime of 250ps, 160% triplet pair yield would imply 320% triplet yield. I guess the authors mean 160% triplet yield, and therefore 80% triplet pair yield.

Response: The reviewer is correct. The standard way to report yields in intramolecular systems is to refer to the total triplet yield (scale of 0 – 200%) rather than the triplet pair yield (0 – 100% scale). Here, 160% means the total triplet yield and the triplet pair yield is 80%. We have clarified this in the revision.

There are a few typos that I have spotted:

"isoindio" in the abstract.

Response: We have corrected these in the revision.

Reviewers' comments:

Reviewer #1 (Remarks to the Author):

I am content with the response of the authors to my comments.

Reviewer #3 (Remarks to the Author):

I am satisfied with the responses and clarifications made to the manuscript.

Reviewer #4 (Remarks to the Author):

This study by Hu et al of intramolecular singlet fission within an isoindigo polymer is an interesting new contribution to the singlet fission field. Potentially important insights into the mechanism of fission in this system go well beyond earlier polymer studies. The experiments appear to have been performed to a high level, and the resulting model is certainly a plausible interpretation. Overall I think the work is strong. However, I believe the authors should consider an additional experiment and another approach to the data analysis (which may help to more comprehensively address the concerns of the other reviewers). Moreover, the manuscript does not always treat the current singlet fission literature appropriately, overlooking several important and likely enlightening opportunities for deeper discussion. My detailed comments are listed below. While there are many, these are all minor concerns that I anticipate can be straightforwardly addressed and would make the study stronger, and that the manuscript would then be well suited to publication in Nature Communications.

--Previous reviews

I consider the authors' response to previous reviewers' comments entirely adequate except for the issue of spectral decomposition, which is nearly satisfactory. The authors' discussion and clarification of the spectral decomposition and meaning of the extracted species is helpful, but I think more directly addressing the 'two-state' model as I suggest below would more conclusively resolve the matter. The presentation of a two-constant fit in the response to Reviewer #1 is instructive, but it doesn't quite address the point raised by the reviewer about overfitting. Similarly, I think the temperature-dependent measurements suggested by Reviewer #3 provided less clarity than hoped, and that the authors should consider routes to isolate the dynamic vs static driving force contributions, perhaps using a rigid host matrix.

--Two-state/three-component model

1) My understanding of this model is that there are only two relevant electronic species represented in the maps in figs 2 and 3, namely S1 and TT. Because they do not exhibit a simple sequential progression but rather have a long period of coexistence, the dynamics are described by three spectral components, the middle one representing the coexistence. This model and the use of these spectral decomposition methods includes the assumption that the spectrum for each electronic species remains constant in time. Is this the case? If so, the authors could provide the S-matrix from singular value decomposition of the datasets, which would clearly demonstrate the presence of two distinct species. Likewise, it should be possible to decompose the datasets in figs 2 and 3 in terms of just the two 'pure' signatures, which would no longer yield exponential time constants but would directly represent the intermediate 'coexistence' phase. This would be a valuable comparison to the traces in Fig 2d, and I would advocate incorporating this complementary representation into the same figure.

2) As in several other instances I highlight in this review, the authors invoke an idea – here an S1 TT coexistence regime – which is plausible but sufficiently controversial that supporting literature should be invoked. For instance, such a model appears in refs 25 (Margulies) and 41 (Stern) as well as Folie et al, JACS 2018.

3) The model becomes unclear when the PL dynamics are discussed. The concept of dynamic TT stabilization is intriguing, but does the assigned TT emission really dynamically redshift? If so, this would give a characteristic S-matrix in the SVD (indicative a very large number of 'species'), but also invalidate the use of the current spectral decomposition techniques. Or can the data be adequately represented as a combination of two time-invariant species? In which case what is the justification for discussion in the text of dynamic changes in the S1-TT offset?

4) The temperature-dependent measurements highlight an interesting mixture between static and dynamic driving-force considerations, but these measurements cannot clearly distinguish between the two. I would suggest attempting to cast the polymers within a rigid host matrix (zeonex?), which may reduce or even switch off the dynamic contribution while leaving the static contribution more or less unchanged. Might this help to clarify the physical picture of the SF process in these polymers? It would also serve to definitively rule out planarization within S1 as the origin of the red-shifted emission.

--Further minor points

5) I interpret the figures on line 91 to refer to the molecular weight and polydispersity of the material, but the units are unclear and there is little context. Much more enlightening was the point made in response to reviewer 3 that the polymers exist in a 'relatively narrow molecular weight distribution'. Can the authors clarify this point in the main text? Are these chains on average long enough that the behavior is unaffected by molecular weight, i.e. are they well beyond the oligomer regime?

6) The sentence in line 59 highlighted by a previous reviewer remains demonstrably false. Ref 15, published two years prior to ref 17, also exhibits intramolecular SF (which is why the authors cite it). iSF was also reported in polydiacetylenes by Lanzani in 1999-2000 and by Kraebel in 1998.

7) The point regarding the need to suppress inter-chain interactions is important, but not entirely new. It is closely related to the well-known design criterion for small-molecule intermolecular fission (see, for example, the 2013 review by Smith and Michl) in which strong intermolecular coupling or a large Davydov splitting can jeopardize fission by stabilizing S1 more than T1. It is a major potential advantage of iSF that the 'inter' problem can be decoupled from fission itself but this does not come out in the introduction.

8) The reference to Dover et al (28) is inappropriate. That work has no relevance to exciton dissociation, but rather the possible role of excimer intermediates in SF. The reference only causes confusion.

9) The authors state on lines 108-110 that they expect the triplet pair state to be less than twice the energy of the free triplet. This is] controversial in the field and must at least be referenced.

10) The reference to Yong et al (37) on line 167 is incorrect: that study is of inter- rather than intramolecular fission.

11) The assignment of red-shifted emission to a directly emissive triplet-pair state is a significant and controversial claim, currently only suggested by three other experimental works (all from the same group: Yong [ref 37], Stern [ref 41] and Lukman JACS 2017). What is the evidence here that the state is direct TT rather than, for instance, a low-energy 'defect'-type state populated by the triplet pair? Because this is an important claim, the authors should more clearly outline their rationale for the assignment and why alternatives can be discarded.

12) In lines 225-7 the authors make a link (or non-link) to the mechanism invoked for TT emission in small-molecule films. The section would benefit from further discussion on how these models are related. Both rely on symmetry breaking to enable electronic coupling between bright and dark states, whether from the shape of the wavefunction or interaction with inter- and/or intramolecular vibrations (which are not treated by the model in ref 26 by the very nature of the approach used). Are the mechanisms truly do different?

13) The statement in lines 304-305 should be moderated: the Bardeen group has demonstrated temperature-dependent singlet fission in single-crystalline tetracene (Piland JPCL 2015).

14) Have the authors investigated the effects of pump laser intensity? Is it possible that intrachain annihilation contributes to the observed dynamics?

Response to the reviewers:

Reviewer #4:

--Two-state/three-component model

1) My understanding of this model is that there are only two relevant electronic species represented in the maps in figs 2 and 3, namely SI and TT. Because they do not exhibit a simple sequential progression but rather have a long period of coexistence, the dynamics are described by three spectral components, the middle one representing the coexistence. This model and the use of these spectral decomposition methods includes the assumption that the spectrum for each electronic species remains constant in time. Is this the case?

Response: The model assumes that the time evolution can be described by a finite number of rate constants such that the extracted spectral components represent the average electronic configuration of the ensemble. This includes the pure singlet at early times and pure triplet pair at late times, with a mix at intermediate times.

If so, the authors could provide the S-matrix from singular value decomposition of the datasets, which would clearly demonstrate the presence of two distinct species. Likewise, it should be possible to decompose the datasets in figs 2 and 3 in terms of just the two 'pure' signatures, which would no longer yield exponential time constants but would directly represent the intermediate 'coexistence' phase. This would be a valuable comparison to the traces in Fig 2d, and I would advocate incorporating this complementary representation into the same figure.

Response: We have determined that the rank of the transient absorption and emission data sets is 3, which directly corresponds to the number of linearly independent components. This is determined using established protocols which are discussed in detail below and which have been added to the SI.^{1,2}

The rank of the data set can be estimated from the number of non-zero singular values as the reviewer suggests. A graphical representation of the singular values (S matrix) is provided below in Figure R2-1a. It is clearly seen by inspection that the first two singular values are above the noise floor and that there are two additional ones that are difficult to distinguish from the noise floor. We have added this graph to the SI but conclude, in agreement with other authors,³ that looking at the S matrix alone is not sufficient to determine the number of components.

Fig R2-1. (a) The singular value decomposition of transient absorption data. (b) The left singular vectors of transient absorption data. (c) The right singular vectors of transient absorption data.

Following this, it is standard protocol to reconstruct the data using an increasing number of components to determine the point when the reconstructed data no longer differs from the original data set by an amount larger than the noise. This analysis was added to the SI in the previous revision and is reproduced here in Figures R2-2e,f. The two component analysis (R2-2c,f) clearly exhibits regions where $\Delta\Delta > 0.5$, which is greater than the noise of the measurement by a considerable margin. In contrast, the three component fit (R2-2b,e) satisfactorily reproduces the data, suggesting we have chosen the appropriate rank=3 for our analysis.

Figure R2-2. (a) Transient absorption of IIDDT-Me in the diluted DCB solution is shown in a pseudo-color plot. (b) The global fitting of three exponentials. (c) The global fitting of two exponentials. (d) The dynamics taken from 700 nm (gray circles) and 810 nm (purple square), and the kinetic traces are shown in two time-constant fits (black line at 700 nm and blue line at 810nm) and three time-constant fits (gray line at 700 nm and purple line at 810 nm), respectively. (e) The residual transient absorption after global fitting of three exponentials. (f) The residual transient absorption after global fitting of two exponentials.

An alternative scheme has suggested that utilizes a visual inspection of the principal kinetics (U) and principal spectra (V) scaled by the square root of the singular values $S^{1/2}$.² This treatment is show in Figure R2-3. Again, we conclude that rank 3 is the

appropriate assignment, since the fourth scaled principal components are not significantly different than the zero lines.

Figure R2-3. Principal vectors scaled by the square root of the singular value. The 4th principal vectors are not distinguishable from the zero line.

Together, these analyses support our use of a rank 3 global analysis method. A similar analysis has been employed for the transient emission data and is found below and in the SI.

2) As in several other instances I highlight in this review, the authors invoke an idea – here an SI TT coexistence regime – which is plausible but sufficiently controversial that supporting literature should be invoked. For instance, such a model appears in refs 25 (Margulies) and 41 (Stern) as well as Folie et al, JACS 2018.

Response: We added the following discussion to the main text that includes a citation to Folie et al. (ref.39):

“The coexistence of S_1 and TT states has been previously observed in molecular systems, such as tetracene, pentacene and terrylenediimide derivatives.^{25, 38-39}”

3) The model becomes unclear when the PL dynamics are discussed. The concept of dynamic TT stabilization is intriguing, but does the assigned TT emission really dynamically redshift? If so, this would give a characteristic S-matrix in the SVD (indicative a very large number of ‘species’), but also invalidate the use of the current

spectral decomposition techniques. Or can the data be adequately represented as a combination of two time-invariant species? In which case what is the justification for discussion in the text of dynamic changes in the SI-TT offset?

Response: The leading terms in the S-matrix and corresponding left and right singular vectors for the TRPL data are shown in Fig. R2-4.

Fig R2-4. (a) The singular value decomposition of transient fluorescence data. (b) The left singular vectors of transient fluorescence data. (c) The right singular vectors of transient fluorescence data.

As we discussed above, the S-matrix is not sufficient to determine the rank of the data set. Instead, we follow a similar analysis (Fig. R2-5), where reconstruct the data using an increasing number of components to determine the point when the reconstructed data no longer differs from the original data set by an amount larger than the noise.

Fig R2-5. (a) The raw TRPL data. (b) Reconstructed TRPL data generated from a 3 exponential global fit. (c) The residual TRPL signal from a 3 exponential sequential global analysis fit.

Again, we find that a rank 3 sequential treatment is appropriate to describe our system. We have made minor modifications to our language to better reflect the fact that the dynamical redshifts in our system are not continuous, but rather seem to reflect a higher and lower triplet pair state on average.

4) The temperature-dependent measurements highlight an interesting mixture between static and dynamic driving-force considerations, but these measurements cannot cleanly distinguish between the two. I would suggest attempting to cast the polymers within a rigid host matrix (zeonex?), which may reduce or even switch off the dynamic contribution while leaving the static contribution more or less unchanged. Might this help to clarify the physical picture of the SF process in these polymers? It would also serve to definitively rule out planarization within S1 as the origin of the red-shifted emission.

Response: We have found that it is not possible to embed the IIDDT-Me polymer in a rigid matrix without inducing structural changes that dramatically redshift the absorption spectra, analogous what occurs in a poor solvent. This effect occurs in both non-polar (e.g., polystyrene) and polar (e.g., polycarbonate and PMMA) matrices. The change in electronic structure can be seen clearly in the UV-Vis spectrum below:

Fig R2-6. Normalized UV-vis absorption spectra of IIDDT-Me solution, IIDDT-Me/PS solution, IIDDT-Me/PC solution, IIDDT-Me film, IIDDT-Me/PS film, and IIDDT-Me/PC film. Chloroform was used as the solvent, and both films were

casted from chloroform solution.

--Further minor points

5) *I interpret the figures on line 91 to refer to the molecular weight and polydispersity of the material, but the units are unclear and there is little context. Much more enlightening was the point made in response to reviewer 3 that the polymers exist in a 'relatively narrow molecular weight distribution'. Can the authors clarify this point in the main text? Are these chains on average long enough that the behavior is unaffected by molecular weight, i.e. are they well beyond the oligomer regime?*

Response: We have clarified this in the main text, which now reads as:

“The isoindigo-based polymer IIDDT-Me ($M_n = 20.1$ kDa, $\mathcal{D} = 3.9$) was synthesized by the copolymerization of electron-deficient isoindigo chromophore and electron-rich bithiophene unit²⁷ and has the chemical structure depicted in Figure 1a. The polymer chains well beyond the oligomer regime as evidenced by the high molecular weight, and previous study has shown that the polymer chains exhibit a relatively narrow molecular weight distribution.²⁷”

6) *The sentence in line 59 highlighted by a previous reviewer remains demonstrably false. Ref 15, published two years prior to ref 17, also exhibits intramolecular SF (which is why the authors cite it). iSF was also reported in polydiacetylenes by Lanzani in 1999-2000 and by Kraabel in 1998.*

Response: “Still, despite being the first reported iSF system,¹⁷” has been changed to: “Still, despite being the first reported iSF system with yields in the multiple exciton generation regime ($> 100\%$),¹⁷ ...”

7) *The point regarding the need to suppress inter-chain interactions is important, but not entirely new. It is closely related to the well-known design criterion for small-molecule intermolecular fission (see, for example, the 2013 review by Smith and Michl) in which strong intermolecular coupling or a large Davydov splitting can jeopardize fission by stabilizing S1 more than T1. It is a major potential advantage of iSF that the 'inter' problem can be decoupled from fission itself but this does not come out in the introduction.*

Response: We have pointed out this in our discussion of aggregation quenching of singlet fission and cited the review paper by Smith and Michl (Ref. 5, *Annu. Rev. Phys. Chem.*, **2013**, 64, 361-386) and Dover et al. (Ref. 47). The discussion now reads:

“Similar concerns about the effects of strong chromophore coupling on the energy and character of the singlet state has been expressed in small molecule systems.^{5,47}”

8) *The reference to Dover et al (28) is inappropriate. That work has no relevance to exciton dissociation, but rather the possible role of excimer intermediates in SF. The reference only causes confusion.*

Response: This reference was added at the suggestion of another reviewer. We have moved it to better reinforce the idea that intermolecular interactions are important design considerations. The discussion now reads:

“Similar concerns about the effects of strong chromophore coupling on the energy and character of the singlet state has been expressed in small molecule systems.^{5,47,}”

9) *The authors state on lines 108-110 that they expect the triplet pair state to be less than twice the energy of the free triplet. This is controversial in the field and must at least be referenced.*

Response: We have added the following citations: Smith, Michl 2010 (Ref. 2), Sanders et al. 2015 (Ref. 20), Aryanpour et al. 2015 (Ref. 32) and Trinh et al. 2017 (Ref. 33).

10) *The reference to Yong et al (37) on line 167 is incorrect: that study is of inter-rather than intramolecular fission.*

Response: Thanks for pointing out this. We have removed this reference and cited another intramolecular fission paper (Sanders, et al. *J. Am. Chem. Soc.* **138**, 7289-7297 (2016)).

11) *The assignment of red-shifted emission to a directly emissive triplet-pair state is a significant and controversial claim, currently only suggested by three other experimental works (all from the same group: Yong [ref 37], Stern [ref 41] and Lukman JACS 2017). What is the evidence here that the state is direct TT rather than, for instance, a low-energy ‘defect’-type state populated by the triplet pair? Because this is an important claim, the authors should more clearly outline their rationale for the assignment and why alternatives can be discarded.*

Response: As these are monodisperse, single polymer chains in solution, we do not expect a large quantity of “defect states,” in contrast to thin film samples, where these sites have been reported to play a non-trivial role in promoting singlet fission. This expectation is reinforced by the magnitude of the stimulated emission signal in TA measurements, which confirms that the behavior is intrinsic to the ensemble and does not result from a small minority of sites. Unlike in TRPL measurements, where minority species can dominate the overall signal, the TA measurements would not reflect defect contributions at the level of a few percent.

12) *In lines 225-7 the authors make a link (or non-link) to the mechanism invoked for*

TT emission in small-molecule films. The section would benefit from further discussion on how these models are related. Both rely on symmetry breaking to enable electronic coupling between bright and dark states, whether from the shape of the wavefunction or interaction with inter- and/or intramolecular vibrations (which are not treated by the model in ref 26 by the very nature of the approach used). Are the mechanisms truly do different?

Response: Unlike polymer iSF materials, there has never been a report of emissive states that are spectrally different from the prompt emission in small molecule iSF systems, nor has direct TT emission been theoretically predicted. So while the idea of direct TT emission is still an open question in small molecule iSF, there is clear evidence for it in polymeric iSF.

13) The statement in lines 304-305 should be moderated: the Bardeen group has demonstrated temperature-dependent singlet fission in single-crystalline tetracene (Piland JPCL 2015).

Response: We have modified this statement and which now reads as: “Unlike **most reported** singlet fission systems,^{38, 40, 44-45} IIDDT-Me exhibits a temperature dependent rate constant for singlet fission.”

14) Have the authors investigated the effects of pump laser intensity? Is it possible that intrachain annihilation contributes to the observed dynamics?

Response: The pump laser intensity is $\sim 75 \mu\text{J}/\text{cm}^2$. We verified that this fluence corresponds to the low fluence regime, where the dynamics are independent of input fluence. We have added the following figure to the SI (Fig S7) to show this:

Figure R2-7. Transient absorption of IIDDT-Me in the DCB solution at (a) $37.5 \mu\text{J}/\text{cm}^2$, (b) $75 \mu\text{J}/\text{cm}^2$, (c) $112.5 \mu\text{J}/\text{cm}^2$ pump intensity are shown in pseudo-colour plots, and the normalized kinetics at 720 nm is shown in (d).

References:

- Henry, E. R. & Hofrichter, J. [8] Singular value decomposition: Application to analysis of experimental data. in *Methods in Enzymology* **210**, 129–192 (Academic Press, 1992).
- DeSa, R. J. & Matheson, I. B. C. A Practical Approach to Interpretation of Singular Value Decomposition Results. in *Methods in Enzymology* **384**, 1–8 (Academic Press, 2004).
- Satzger, H. & Zinth, W. Visualization of transient absorption dynamics – towards a qualitative view of complex reaction kinetics. *Chem. Phys.* **295**, 287–295 (2003).

REVIEWERS' COMMENTS:

Reviewer #4 (Remarks to the Author):

The authors' detailed presentation of the spectral decomposition methods is appreciated. The clarifications added help to address my concerns and those raised by previous reviewers. It is unfortunate that the polymer cannot be studied in rigid matrix or at low temperature without significant redshifts, but I am satisfied that the authors made the effort.

I maintain the point I made in item 12 of my previous review, that there are intriguing similarities in the models invoked for direct TT emission in the solid state in small-molecule films and the mechanism invoked here, and that these merit discussion. The authors' response focuses on small-molecule iSF systems, which I did not ask about. The previous reports I named from the Friend group all show 'emissive states that are spectrally different from the prompt emission' following intermolecular singlet fission, just as is reported here in iSF. This does not detract from the clear novelty of the current findings. Given the phenomenon of emissive TT states is evidently of growing interest in the field, I think the authors should discuss this aspect further.

This minor point notwithstanding, I am in favor of publication of the work.

Response to the reviewers:

Reviewer #4:

I maintain the point I made in item 12 of my previous review, that there are intriguing similarities in the models invoked for direct TT emission in the solid state in small-molecule films and the mechanism invoked here, and that these merit discussion. The authors' response focuses on small-molecule iSF systems, which I did not ask about. The previous reports I named from the Friend group all show 'emissive states that are spectrally different from the prompt emission' following intermolecular singlet fission, just as is reported here in iSF. This does not detract from the clear novelty of the current findings. Given the phenomenon of emissive TT states is evidently of growing interest in the field, I think the authors should discuss this aspect further.

This minor point notwithstanding, I am in favor of publication of the work.

Response: As suggested, we have identified symmetry breaking as the common theme in these two systems. We have included an additional citation from the Cambridge group by Thampi et al. (Ref. 40, J. Am. Chem. Soc. 2018, 140, 4613) that also reports TT emission in TIPS-tetracene nanoparticle aggregates. We modified the final revision as follows:

“.....We note that a similar phenomenon has recently been reported in molecular crystals, where a distinct red-shifted delayed emission signal was observed and assigned to direct TT emission.⁴⁰⁻⁴¹ While the prediction of direct TT emission in polymers is based on purely electronic considerations, and doesn't involve phonon-mediated intensity borrowing arguments that have recently been invoked in molecular crystals,⁴⁰ both reports suggest that symmetry breaking is a necessary condition for the observation of direct TT emission.”